# What Influences Farmers’ Adoption of Soil Testing and Formulated Fertilization Technology in Black Soil Areas? An Empirical Analysis Based on Logistic-ISM Model

**DOI:** 10.3390/ijerph192315682

**Published:** 2022-11-25

**Authors:** Yuxuan Xu, Hongbin Liu, Jie Lyu, Ying Xue

**Affiliations:** 1College of Economics and Management, Shenyang Agricultural University, Shenyang 110866, China; 2College of Land and Environment, Shenyang Agricultural University, Shenyang 110866, China

**Keywords:** soil testing and formulated fertilization technology, technology adoption behavior, logistic-ISM model, black soil ecosystem conservation

## Abstract

Along with the increasing prominence of environmental risks such as soil surface source pollution and declining quality grade of arable land, the issues of how to address irrational fertilizer application and enhance the safety of agricultural products have attracted widespread attention. In this context, clarifying the main factors affecting farmers’ use of soil testing and formulated fertilization technology (STFFT) can further improve the technology adoption rate and fertilizer utilization efficiency, promote standardized agricultural production and maintain the health and stability of soil ecology in black soil areas. This is of great significance to the construction of green agriculture, national dietary health and national food security. This study builds an “external environmental stimuli-perceived characteristics-adoption behavior” theoretical framework to investigate the decision-making and the dynamic influence mechanisms of farmers’ adoption behavior of STFFT. Based on farmer survey data, the logistic-ISM model has been applied. The main findings are as follows. First, five types of influencing factors, namely individual characteristics, family characteristics, business characteristics, cognitive characteristics and external environmental characteristics, had significant “push” effects on farmers’ STFFT adoption behavior. Among them, planting scale and technical training are the key factors influencing farmers’ adoption of scientific fertilizer application technology. Second, both farmers’ perceived ease of use and perceived usefulness play a significant role in farmers’ decision-making process, and the easier farmers perceive STFFT to be to master and the greater the benefits it brings, the more pronounced the tendency to adopt the technology, all other influencing conditions being equal. Third, the main influencing factors of farmers’ STFFT adoption behavior are intrinsically related and divided into four categories based on the magnitude of influence: deep-rooted, medium indirect, shallow indirect and superficial direct. In order to reduce further degradation of black soil caused by farmers’ irrational production habits and to improve resource utilization efficiency, this study recommends the government to further regulate the land transfer market, strengthen the propagation of soil-conservation-type technologies in black soil areas, expand the breadth of agricultural technology training and enhance farmers’ understanding and trust in STFFT. Thus, the maintenance of soil ecosystem in black soil areas, effective guarantee of food security and sustainable development of agriculture can be achieved.

## 1. Introduction

Under the influence of unstable factors such as frequent global extreme weather events and complex changes in the supply chain of agricultural products, how to increase food yield and improve agricultural production efficiency is an important global issue to ensure regional food security and social stability [1,2,3]. As a basic industry, agriculture has assumed an important role in maintaining social harmony and healthy development of the people under the pressure of China’s large base population. In the context of the gradual weakening of the urban–rural dichotomy, the farmer group is constrained by both human capital and natural conditions. Under this circumstance, to better promote food security and guarantee the self-sufficiency rate, farmers often stimulate the increase in agricultural production by raising the application of chemical fertilizers [4,5]. This unreasonable fertilizer input behavior can make farmers obviously observe the increase in agricultural products yield in the short term, but the long-term application of this method will cause a series of consequences such as soil slabbing, erosion and salinization, which will further deteriorate the soil environment, destroy the stability of the ecosystem and expand agricultural surface pollution [6,7]. According to available studies, the internationally accepted upper limit of environmentally safe fertilizer application is 225 kg/ha, while the current level of fertilizer application in China is much higher than this value [8]. This process creates a vicious cycle of “resource constraint-over fertilization–ecological deterioration–intensification of constraint”, which poses a stronger challenge to a series of actions taken by China to ensure food security. Therefore, how to balance economic and ecological benefits in the agricultural production chain is a question that needs urgent in-depth consideration. As the most important production factor in the process of food cultivation, land resources are one of the key factors to guarantee the structural integrity of the agroecological system, and their conservation and sustainable use is an important foundation for national economic development and social harmony and stability [9,10,11]. As one of the rare soil resources, black soil is an object of urgent attention and protection [12]. On this basis, the key issue that needs to be solved now is how to scientifically and rationally analyze the problems faced by land resources in the process of agricultural production, alleviate the unbalanced relationship between “economy–ecology”, promote the sustainable development of soil ecosystems and enhance the feedback capacity between “people–land” feedback capability.

To further enhance the efficiency of fertilizer utilization and improve the soil quality level and ecological environment of farmland, STFFT is gradually being promoted and accepted [13,14]. The proliferation of this technology has promoted agricultural carbon peaking and carbon neutrality [15]. STFFT is a pro-environmental type of soil conservation tool, which is a scientific fertilizer application technique designed to address the current soil deterioration problems caused by the overuse of chemical fertilizers [16,17]. Specifically, STFFT refers to an agricultural technology based on soil testing and fertilizer field trials to determine the amount, period of use and working method of elements such as nitrogen, phosphorus, potassium, micronutrients and organic matter based on the pattern of crop demand for elements, the ability of the soil to supply fertilizer and the effect of fertilizer application [18,19]. The implementation process of the technology includes five continuous working links, “soil determination-formulation–fertilizer formulation–fertilizer supply–fertilizer application guidance”, with the first three links as the main steps to systematically provide fertilizer application guidance services for farmers. In this regard, soil determination means that after sampling the soil in the region, the nutrient indicators of the soil are learned through test identification. After understanding the soil’s ability to provide fertilizer, the testing facility transitions to the formulation process, which ultimately results in a fertilizer recommendation card that can provide guidance to farmers. Farmers can choose to use a comprehensive mix of individual fertilizers based on recommendations, or they can purchase specific fertilizers produced by companies that obtain soil tests (with regional variations). STFFT achieves the two most obvious advantages of scientific quantification and symptomatic treatment by mastering the nutrient demand of crops and the nutrient supply of the soil [20,21,22,23]. This technology is rooted in the basic theories of the three laws of fertilizer, the doctrine of nutrient restitution and the law of diminishing returns, which effectively reduces waste of resources and improves the efficiency of fertilizer utilization, thus ensuring the ecological balance and structural integrity of the soil [24]. STFFT can aptly address the supply-and-demand relationship between crop growth and fertilizer inputs, determine the most scientific and reasonable fertilizer application rate to achieve the best dynamic balance point of high yield and low cost and use it as a basis to achieve the comprehensive purpose of protecting the ecological environment. However, the extent of STFFT diffusion is still lacking, and the adoption rate of farmers is relatively low. Farmers are the main actors in agricultural production behavior, technology adoption, land use and participation in the soil ecosystem. Therefore, a systematic analysis of farmers’ STFFT adoption cannot be ignored.

Research on soil testing, fertilizer application and agricultural technology adoption behavior has been conducted by more scholars around aspects related to farmers’ decision-making mechanisms and influencing factors [25,26,27,28,29,30,31]. Much of the existing research on soil testing and fertilization has centered on its significance and impact. For example, Shasha took an in-depth look at the current situation of STFFT promotion in China and pointed out a series of shortcomings such as the outstanding contradiction between supply and demand in soil testing, lack of supervision in fertilizer distribution and inadequate services in fertilizer application [32]. Using sampling data from Nankang District, Jiangxi Province, Shu analyzed in depth the benefits that can be brought by using STFFT from the perspective of field shape, area and water flow direction [33]. Studies on agricultural technology adoption behavior have mostly been analyzed by scholars in terms of intervening factors and late effects of technology adoption [34,35,36,37]. Zhang et al. used a matching method to examine the facilitative role of cooperatives in agricultural technology adoption, particularly the effect on the breadth of adoption, based on farmer research data from Sichuan Province, China [38]. Manda et al. similarly analyzed the impact of cooperative membership in farmer technology adoption using data from two rounds of surveys that occurred in maize and soybean growing areas in eastern Zambia between 2012 and 2015. Unlike Zhang, this study concluded that cooperative membership positively influenced technology adoption for fertilizer application [39]. Wang conducted a survey and analysis based on the Minqin area in the northern part of the Hexi Corridor in China. He empirically tested the contribution of social capital and service diffusion to farmers’ technology adoption efficiency through a stochastic frontier model [40]. The subsequent impact of technology adoption was of interest to Wossen et al., who used data from a sample of cassava farmers in Nigeria to further clarify the impact of technology adoption on farmers’ family welfare levels and showed that the effect of technology diffusion and adoption on poverty reduction was positive [41]. In general, existing studies have examined farmers’ technology adoption behavior in some depth, but there are still some areas for further improvement as follows. First, the research method is single. Existing studies mostly use a single statistical-econometric model for analysis, which still has limitations in the interpretation of the research results. This study further combines the advantages of the ISM model of association hierarchy mining on the basis of the logistic regression analysis method to make the research results more dimensional. Second, the research content still needs to be improved. The research on STFFT effects and adoption behavior is mostly focused on rice cultivation and vegetable cultivation, but there is less research on STFFT adoption behavior of other crops, and the research on the main factors affecting farmers’ STFFT adoption lacks deeper discussion and has room for further research expansion. Therefore, this study examines the main factors influencing farmers’ STFFT adoption in terms of their basic characteristics, cognitive characteristics and external interventions, based on technology acceptance theory.

In summary, STFFT adoption by farmers, as an important behavior in the agricultural production process, has an important impact on the ecological safety protection and physicochemical property enhancement of soil. Black soil is one of the most fertile and precious land resources in the world. The black soil area located in northeastern China has a total area of approximately 1.09 million square kilometers, which is one of the three largest black soil areas in the world. However, under the action of natural factors such as erosion, slabbing and salinization, this land has also experienced long-term exploitation by high-intensity rough farming activities, which has led to a gradual loss of organic matter content in black soils, changes in the area and structure of soils and decreases in soil consolidation, water retention and fertilization capacity in varying proportions, as well as decreases in ecological and economic benefits [42,43,44]. Liaoning Province is an important black soil distribution area in China and also a pilot province for STFFT promotion. In-depth investigation of the factors influencing farmers’ adoption of STFFT in this area not only has an important impact on soil system conservation, but also plays an important role in the improvement of eco-efficiency and long-term stable development of agriculture. Therefore, based on technology acceptance theory, this study constructs a theoretical analysis framework of “external environmental stimulus–perceived characteristics–adoption behavior” for farmers’ use of STFFT, aiming to clarify the factors influencing farmers’ decision-making and their influence paths. This study used field interview data from 402 maize farmers in Liaoning Province to empirically test the theoretical analysis framework using a logistic-ISM model. In this model, which has both qualitative and quantitative analytical capabilities, farmers’ decisions are considered to be influenced by a combination of individual characteristics, family characteristics, business characteristics, cognitive characteristics and external environmental characteristics, while the influences form interconnected paths in the process. The study may have two contributions. Theoretically, it can more accurately express the behavioral decision-making process of small-scale farmers and provide new research ideas, analytical frameworks and methodological systems for the application of technology acceptance theory and the thesis of how to promote a stable balance in soil ecology. In practice, this study can provide a realistic basis for stimulating farmers’ enthusiasm to understand STFFT, increasing the adoption rate of STFFT and realizing the structural reform of agricultural supply side and environment-friendly production at the micro level. At the macro level, this study can help government departments scientifically establish STFFT service systems and service mechanisms that meet farmers’ needs and increase the adoption rate of green agricultural production technologies, thus promoting sustainable agricultural development and helping the implementation of the rural revitalization strategy.

## 2. Theoretical Analysis

Technology acceptance theory was developed by Davis in the late 1980s by synthesizing behavioral theories, and the most basic idea of the model is that the actor’s intention to use a technology determines the final behavior. In this theory, intention to use is determined by both perceived usefulness and ease of use of the technology as perceived by the decision subject. Among them, perceived ease of use has an impact on perceived usefulness when stimulated by external variables [45,46,47]. It is worth noting that this theoretical model should be closely integrated with the real environment when applied. Different assumptions should be made for the heterogeneity of influencing factors so as to ensure the accuracy of the model and, thus, scientifically explain farmers’ technology adoption behavior [48]. With subsequent revisions and extensions by scholars, the theory has been widely applied in the areas of sharing economy, intelligent transportation systems and agricultural technology [49,50,51,52,53,54]. Shamsi analyzed the well-being of 3140 academic staff at three Norwegian universities, using technology acceptance theory to delve further into the impact of using online forms of teleconferencing in daily teaching and other tasks on well-being in the context of the global new coronary pneumonia [55]. Wei et al. constructed an intentional influence model based on the technology acceptance model and perceived risk theory, mainly devoted to the study of consumers’ propensity to make fruit purchases online, and an online survey of 344 consumers under 30 years of age was used to test the theoretical model. The results showed that online discounts on fruit significantly increase consumers’ purchase interest and stimulate a willingness to buy online [56]. Existing research has achieved many valuable academic results using technology acceptance theory, which has laid a solid foundation for this paper on farmers’ STFFT adoption. Based on this foundation, this study constructs a theoretical analysis framework for farmers’ STFFT adoption covering the joint influence of external stimuli, basic features, perceived ease of use and perceived usefulness (Figure 1).

STFFT is a new agricultural technology that guides farmers to apply fertilizer rationally, and whether or not they will adopt the technology is determined by farmers’ intention to use it, which depends on their judgment of the effectiveness and operability of the technology. The ease of mastering a technology affects the perceived usefulness of the farmer, in other words, the easier a technology is to master, the stronger the effect of its use perceived by the farmer. This is because, when farmers are not confident in mastering a technology, they also remain confused and skeptical about the effectiveness of its use. In addition, farmers, as limited rational actors, are economically motivated by the expected profit maximization in their technology adoption behavior and clarifying the relationship between the expected benefits of technology adoption and technology adoption behavior is the premise and basis for studying farmers’ STFFT adoption behavior. When farmers choose a production technology in agricultural production, the expected benefits of technology adoption are the key factors influencing their choice decision, and the composition of total income and total costs resulting from technology adoption directly affects the expected benefits of technology adoption. Farmers will think rationally and make a thorough comparison in order to achieve the goal of maximizing their expected profit. On the one hand, farmers will choose technologies that are conducive to improving the quality of their agricultural products for better purchase prices in the market and increasing their total income. On the other hand, they will choose technologies that are conducive to increasing the yield of agricultural products, and under the constraints of existing environmental conditions, how to use limited natural resources to obtain more agricultural products is a common goal pursued by farmers. In addition, technology adoption costs are reduced and the difficulty of adopting the technology is reduced so that farmers will accept it financially and psychologically, gain more profit and thus maximize profit. Therefore, measuring farmers’ perceived usefulness and perceived ease of use of STFFT is an important way to study farmers’ willingness to adopt technology.

Due to the large uncertainty in agricultural production, in addition to the influence of psychological factors, farmers are also influenced by other factors such as their family characteristics and natural environmental characteristics in the decision-making process of production activities. Whether STFFT can be recognized and adopted by farmers requires a combination of factors influencing farmer-related characteristics, in addition to the characteristics of the technology itself. First, government publicity, government subsidies, organizational participation, technology training and land transfer together constitute the external environmental factors that influence farmers’ judgment of the expected profit level after adopting STFFT, which in turn affects farmers’ adoption behavior decisions. Among them, the more the government and organizations publicize and promote STFFT, the better farmers’ understanding of the technology and the more farmers’ willingness to adopt the technology is stimulated. As rational actors, the adoption of new technologies often relies on a strong risk uncertainty, and farmers maintain a wait-and-see attitude in the early stage. The government, as an organization with higher credibility, has a non-negligible role in dispelling farmers’ concerns and promoting farmers’ adoption of STFFT technologies. Second, farmers’ cognitive ability and basic experience level vary due to their gender, age, education level, information access, family income structure, part-time employment, planting scale and planting time. This leads to a differential impact on farmers’ cognitive level, perception, comprehension and learning ability when faced with new production technologies. As a result, farmers’ judgments of expecting profits are also more disparate. These differences will in turn affect farmers’ adoption behavior decisions, resulting in very different levels of technology adoption for the same level of technology diffusion. In summary, farmers’ STFFT adoption behavior is a complex process that is influenced by both farmers’ own internal factors and stimulated by external environmental factors. In order to investigate the inner mechanism of farmers’ STFFT adoption behavior in depth, it is necessary to systematically sort out and comprehensively summarize the relevant influencing factors.

## 3. Materials and Methods

### 3.1. Research Area

The arable land in the black soil area of Liaoning Province accounts for 10.07% of the arable land area in the typical black soil area of northeastern China and is an important commercial production food base in northeast China [57,58]. However, with the long-term excessive application of chemical fertilizers, a series of negative effects have gradually emerged, posing a serious threat to the green development of agriculture [59]. In the past 30 years, the organic matter content of the cultivated layer of black soil in Liaoning Province has decreased by nearly 16% and the thickness has decreased by approximately 13 cm (Source: Green Agriculture Technology Center of Liaoning Province). In this context, how to protect the black soil has become an urgent problem to be solved [60,61]. Under such objective reality, the Liaoning Provincial Government attaches great importance to the governance of chemical fertilizer application and has released relevant documents such as “Work Specification for Soil Testing and Fertilizer Application in Liaoning Province” and “Basic Requirements for the Construction of County Laboratories for Soil Testing and Fertilizer Application in Liaoning Province” in succession and has promoted STFFT since 2005. The work progress so far: more than 32,000 soil samples have been collected in the province, more than 128,000 sets of effective experimental data have been obtained, more than 275,000 copies of soil formula fertilizer recommendation cards have been issued, the cumulative area of soil formula fertilizer measurement has been promoted more than 65 million mu times and the amount of formula fertilizer applied has reached more than 1.75 million tons, exceeding the total amount of base-applied chemical fertilizer by more than 35% (Liaoning Statistical Yearbook 2017). At present, Liaoning Province has established the STFFT system of “taking soil in the field → laboratory analysis → field test → formulation → issuance of soil testing formula fertilizer recommendation card → fertilizer distribution → field guidance → effect feedback”. Considering that maize cultivation area accounts for approximately 3/5 of the total crop cultivation area in Liaoning Province (Source: Liaoning Province Statistical Yearbook 2021), maize growers were selected as the respondents of this study. The research area designed for this study was eight counties and cities under four directions in Liaoning Province, which basically covered the main maize producing areas in the province and could reflect the maize planting production in the province in a more comprehensive way (Figure 2).

### 3.2. Data Sources

The data used in this study were derived from a questionnaire survey of maize farmers in eight counties and cities in Liaoning Province conducted by the group from May to July 2018. The questionnaire was generally divided into seven sections, which mainly included information on farmers’ basic characteristics, family production and operation, planting, fertilization, technology adoption, information capacity, social capital situation and external environmental characteristics. The sampling process designed for this study is as follows. First, this study divided Liaoning province into four regions geographically based on the orientation of space. Second, two administrative counties (cities) were randomly selected from within each geographic region. Third, 2–3 administrative villages were selected from within each surveyed county (city) based on the principle of randomness. Using this as the base study area, farmers were randomly researched. The team first conducted a small-scale pre-survey in Hengren County to ensure the rationality and rigor of the questionnaire. For meeting the authenticity and validity of the survey results, the investigators were trained in statistical methods and language expressions and other related aspects. The actual survey was conducted in the way of farmers’ dictation and surveyors filling out the questionnaire on site, which ensured the integrity of the data to a certain extent. A total of 427 questionnaires were collected, and 402 valid questionnaires were obtained by eliminating invalid questionnaires, with an effective rate of 94.15%. Among them, there were 99 copies in the eastern region, 96 copies in the southern region, 99 copies in the western region and 108 copies in the northern region. Due to the different number of maize farmers in the surveyed administrative villages, the number of samples obtained in each survey area was different, but the survey data were generally distributed more evenly.

### 3.3. Research Methodology

To further test the theoretical logical framework of “external environmental stimuli–perceived characteristics–adoption behavior”, this study adopts a combination of binary logistic model and interpretative structural model (ISM). Such a combination breaks through the limitations of using quantitative or qualitative analysis alone. After using regression analysis to obtain the direction and extent of the factors that influence farmers’ adoption of STFFT, the interrelationships and hierarchical structure between the factors can be further obtained through ISM analysis. Compared with existing studies using models such as simple linear and structural equations [14,62], this study breaks through the limitations of quantitative analysis and combines qualitative with it. This allows for a clearer formation of a stepwise progression of relationships and a clarification of the intrinsic connections between elements. Although this combination of logistic-ISM has the advantage of analytical channel expansion, it has some limitations in the duality of dependent variables and the capacity of the factor set. When the number of elements is too large, the output of such complex matrix algorithms and graphs makes the computing time grow exponentially. Currently, more and more scholars are paying attention to the ingenuity of this “combination” and applying it to multidisciplinary fields [63]. For example, Zhang et al. used a combined logistic-ISM approach to investigate, in depth, the divergence between farmers’ willingness to adopt land transfer and their actual behavior within poor mountainous areas in China. The results of the study indicated that factors such as motivation of agricultural production, cultivation scale and farmers’ part-time income had direct or indirect effects on the divergence of their willingness and behavior [64]. In this study, after taking into account the complexity of farmers’ decision-making systems and influence pathways, a comprehensive analysis was conducted using a logistic-ISM model to enhance the scientific validity and reliability of the study.

#### 3.3.1. Binary Logistic Model

This part of the econometric analysis uses a logistic model to explore farmers’ STFFT adoption behavior, with both “yes” and “no” endpoints. It is suitable for the analysis of choice behavior according to the utility maximization principle. The basic form of the logistic model is as follows:(1)$$P=F\left(Y\right)=\frac{1}{1+e^{Y}}$$
in Equation (1), the dependent variable *Y* is whether the farmer adopts STFFT or not (i.e., it is a 0–1 type dependent variable). If the farmer adopts STFFT, then *Y* = 1; if the farmer does not adopt STFFT, then *Y* = 0. The probability of *Y* = 1 is set as *P*. *x_i_* (*i* = 1, 2, …, n) is the explanatory variable, which includes the influencing factors in the basic characteristics, family characteristics, business characteristics, cognitive characteristics and external environmental characteristics of the farmer. Additionally, *Y* is a linear combination of variables *x*_1_, *x*_2_, …, and *x*_n_, which are:(2)$$Y=b_{0}+b_{1}x_{1}+b_{2}x_{2}+\ldots \ldots +b_{n}x_{n}$$
in Equation (2), *b_i_* (*i* = 1, 2, ……, n) is the regression coefficient of the *i*th influencing factor, and if *b_i_* is positive, it means that the *i*th influencing factor has a positive influence on farmers’ adoption of STFFT, and if *b_i_* is negative, it means that the *i*th influencing factor has a negative influence on farmers’ adoption of STFFT. Transformation according to Equations (1) and (2) yields a logistic model in the form of incidence ratio (odds). In Equation (3), *b*_0_ is the constant term and *ε* is the random error.
(3)$$Ln(\frac{P}{1-P})=b_{0}+b_{1}x_{1}+b_{2}x_{2}+\ldots \ldots +b_{n}x_{n}+\varepsilon$$

#### 3.3.2. Interpretation Structure Model

This study takes a systems theory perspective and uses the ISM model to explore, in depth, the intrinsic correlation and hierarchical structure of the influencing factors. This will provide a more direct and specific scientific reference for enhancing farmers’ adoption of STFFT. The ISM model, based on qualitative analysis, can use directional line segments to connect and combine all elements into a system model of an ensemble. This feature can not only transform the intersecting and ambiguous system elements into a more intuitive structural relationship model, but also verify the rationality of the elements. Thus, it can realize the accurate analysis of the problem essence and lay a more solid theoretical foundation for proposing countermeasures. At present, the ISM model has been widely used in many aspects such as farmers’ behavior, supply chain analysis, energy consumption, industrial development and group individual decision making. The intuitiveness, rigor, comprehensiveness and scientificity of the model’s analysis results have been generally recognized by the academic community [65,66,67]. Combined with the actual situation, an ISM operation group was formed in this study. The team consisted of nine people, including research scholars, provincial STFFT extension leaders, project leaders in the surveyed areas, leaders of agricultural cooperatives and doctoral and master’s students. All members of the operation team have more in-depth research on farmers’ STFFT adoption behavior and have a broader perspective, which lays a solid foundation for the generalizability of the conclusions obtained.

A system in the ISM model is composed of multiple elements that play a direct or indirect role in accordance with certain supporting or inhibiting logical relationships. Therefore, when constructing or transforming a system, it is important to first clarify the elements that make up the system and the logical relationships that exist, which helps to ensure the structure, hierarchy and openness of the system. Suppose *S*_0_ is the farmer STFFT adoption behavior and *S_i_* (*i* = 1, 2, 3, … n) as a set of influencing factors that have an effect on the farmers’ technology adoption behavior is used to construct the adjacency matrix. The adjacency matrix is a basic representation of the factor relationship matrix, which is characterized by a clear description of the direct relationships among the factors. The direct relationships among the factors influencing farmers’ adoption of STFFT are specified in the adjacency matrix A. The element *a_ij_* of the adjacency matrix A can be defined as:(4)$$a_{ij}={\{}_{0,\;\underset{}{}S_{i}\underset{}{}\;and\;\underset{}{}\;S_{j}\;\underset{}{}\;are\;\underset{}{}\;not\underset{}{}\;related\;}^{1,\;\underset{}{}\;S_{i}\;\underset{}{}and\;\underset{}{}\;S_{j}\;\underset{}{}\;related}$$

Let the elements of the *i*th row and *j*th column in matrix A be a. If *a_ij_* = 1, it means that the element *i* that influences the farmer to adopt STFFT has influence on the element j that influences the farmer to adopt STFFT; if *a_ij_* = 0, it means that the element *i* that influences the farmer to adopt STFFT has no influence on the element *j* that influences the farmer to adopt STFFT. Additionally, this is used as the basis for constructing the reachable matrix, which must follow the Boolean algebraic operation, if (A + I) × n = (A + I) × n + 1, then R = (A + I) × n.

#### 3.3.3. Variable Selection

(1)Individual characteristics. Individual characteristics mainly include gender (IC1), age (IC2), education level (IC3) and information acquisition ability (IC4). Theoretically, men are more willing to accept and try new things than women, and they also expect more economic benefits, so men are more willing to adopt STFFT. As farmers grow older, their ideology becomes more conservative, their attitude and behavior become more stable and they are not keen on high-input and more complicated STFFT, so age has a negative impact on farmers’ adoption of STFFT. The higher the education level of farmers affects their ideology and cognitive level, the more educated they are, the more correct their business philosophy, the more standardized their production behavior and the better their ability to understand and apply STFFT, so education level has a positive effect on farmers’ adoption of STFFT. Information is something used to eliminate random uncertainty. The stronger the information acquisition ability, the better the quantity and quality of information obtained, the higher the degree of awareness of the relationship between planting production and ecological environment and the more accurate the judgment of the expected effect of technology adoption, which is conducive to the transmission and sharing of technology, so information acquisition ability has a positive influence on farmers’ adoption of STFFT.(2)Family characteristics. Family characteristics mainly include the proportion of planting income to total family income (FC1) and the proportion of laborers working outside the home (FC2). The main purpose of farmers’ plantation production is to obtain economic benefits. The larger the proportion of plantation income to total family income, the greater the farmers’ dependence on plantation production, the higher the degree of attention to plantation production and the better the understanding of plantation production technology, so the proportion of plantation income to total family income has a positive influence on farmers’ adoption of STFFT. The larger the proportion of laborers working outside the home, the less likely it is that farming will become the main family business, the less attention to farming and the lower the willingness to adopt STFFT, so the proportion of laborers working outside the home has a negative effect on farmers’ adoption of STFFT.(3)Operation characteristics. Operating characteristics mainly include planting scale (BC1) and planting time (BC2). Generally speaking, the larger the farmers’ planting scale, the stronger their management and technology application capabilities, the higher the degree of production intensification and standardization, the greater the production risk they bear and the more inclined they are to adopt STFFT to increase planting returns, so the planting scale has a positive impact on farmers’ adoption of STFFT. The longer farmers have been engaged in planting, the higher their knowledge of planting production, the better their understanding of the relationship between technology application and planting management and the more inclined they are to adopt STFFT, so planting time has a positive effect on farmers’ adoption of STFFT.(4)Cognitive characteristics. The cognitive characteristics mainly included farmers’ perception of STFFT (PC1), perceived land fertility (PC2), perceived production cost (PC3), perceived ease of use of STFFT (PC4) and perceived usefulness of STFFT (PC5). Therefore, we selected “your perception of STFFT”, “your perception of fertility of all your plots”, “you think the cost of STFFT is lower than the cost of common fertilizer application”, “You can master and apply STFFT faster and better” and “You think STFFT is more useful for planting production” as research variables for the cognitive characteristics.(5)External environment characteristics. The external environment features mainly include government propaganda (SC1), government subsidies (SC2), organizational participation (SC3), technical training (SC4) and land transfer (SC5). Theoretically, government propaganda plays an important role in farmers’ adoption of STFFT, and more and more farmers are willing to respond to the government’s call to actively carry out fertilizer reduction and efficiency improvement, in which STFFT plays an important role in fertilizer reduction and efficiency improvement, so government propaganda has a positive impact on farmers’ adoption of STFFT. Government subsidies can, to a certain extent, reduce the cost of STFFT adoption by farmers and increase farmers’ motivation to adopt it, so government subsidies have a positive impact on farmers’ adoption of STFFT. Organizational participation refers to farmers’ participation in industrial cooperative organizations such as cooperatives, associations and companies. Farmers can purchase lower-priced and quality-assured agricultural materials from industrial cooperative organizations such as cooperatives, associations and companies, and can also obtain scientific planting techniques. Technical training not only enables farmers to master more useful planting production techniques, but also enables farmers to better understand the relationship between fertilizer use and crop yield and change unreasonable fertilizer application behaviors, thus lowering the threshold for STFFT adoption, so technical training has a positive impact on farmers’ adoption of STFFT. STFFT can improve soil fertility, and farmers who have transferred their land are more likely to adopt STFFT to improve soil fertility and increase their farming income, so land transfer has a positive effect on farmers’ adoption of STFFT. Comprehensive analysis of the above, the following variable table was constructed in this study (Table 1).

## 4. Results

### 4.1. Descriptive Statistical Analysis

In this section, we first discuss the descriptive statistics of the variables included in the binary logistic regression model and the ISM model. Then, descriptive statistics are analyzed for the adoption of STFFT in the sample of farmers surveyed in this study. The analysis in this section aims to provide a deeper understanding of the importance of the main phenomenon in the agricultural production area examined in this study, namely STFFT adoption behavior, and the relevant background information that can be used to interpret the results of the descriptive statistical analysis.

A summary of the variables included in the binary logistic regression model and the explanatory structural model in this study is shown by the statistics in Table 1. Of the 402 samples data, 233 farmers adopted STFFT technology, accounting for 57.96%, which is above the half level. However, the level of adoption was not high overall. On average, the majority of farmers in the sample data were male, with 73.70% of the male group using STFFT in their agricultural operations. From the perspective of age, the largest number of sample farmers is between the ages of 46 and 55, but the STFFT adoption rate was much lower than that of farmers whose ages were between 26 and 35. Most of the sample farmers had low education levels, and despite the completion of universal nine-year compulsory education, primary education was still predominant among the farmer group due to the constraints of old-fashioned thinking and economic conditions. In the sample, the number of farmers with an education level of elementary school and below was 247, of which 113 adopted technologies, with a technology adoption rate of 45.75%. This value is much lower than the group of farmers with a high school or secondary school education, 95.24% of whom adopted STFFT. This is due to the fact that farmers with a higher education level have higher knowledge acceptance and risk resistance than the farmers with a lower education level, which has a positive effect on the adoption of STFFT.

To further understand the contextual factors affecting STFFT penetration and adoption, this study conducted a descriptive statistical analysis based on two aspects: the penetration channel of STFFT and the factors prioritized in farmers’ technology adoption decisions. According to Figure 3, most farmers have narrow access to STFFT, and 28.61% of the sample farmers learned about this new fertilizer application technology through technical training. Other channels including social networks, field services of agricultural technicians and modern media also bring new agricultural information to farmers to some extent. However, limited by the information platform in rural areas and the basic literacy of farmers, the technology diffusion channels, mainly modern communication technology, failed to show a significant effect. It is easy to see from the study data that farmers are mainly driven by the profit maximization goal when considering whether to adopt STFFT, with 35.82% of farmers making their decision based on the yield-increasing effect of the new technology. The better the yield increase effect and the higher the expected profit, the more obvious the farmers’ motivation to adopt STFFT. Another 18.41% and 16.67% of farmers made their decisions based on two factors, namely, planting area and learning difficulty of the technology. The better the scale effect and the lower the learning difficulty, the more farmers were inclined to accept STFFT.

### 4.2. Logistic Regression

In this study, the data were regressed using the econometric analysis software Stata 11.0 in order to further clarify the factors influencing farmers’ choice of STFFT. The underlying regression analysis involved the application of a discrete choice model in which the dependent variable was the farmer’s adoption of STFFT. This variable is binary and has a value of 1 if the farmer adopts STFFT and 0 otherwise. Table 2 shows the estimation results of the logistic regression model, where a series of indicators are used to measure the degree of model fit. The values of LR chi^2^ for model 1 and model 2 were 312.20 and 304.72, respectively, and the models had strong explanatory power for the observed variables, and both models were statistically significant. Both models have *p*-values well below the standard 0.05 level (*p* = 0.000) when compared to the chi-square distribution with one degree of freedom, thus indicating the consistency of the meanings expressed by model 1 and model 2. Therefore, we have accurate and strong evidence to reject the original homogeneity hypothesis and accept the alternative hypothesis that heteroscedasticity is actually present in the residuals of our model of this regression. As showed in Table 2, the explanatory variables used in the model are significant for the resulting equation. The regression process at this stage is as follows. First, all variables are introduced into model 1 for estimation. Second, based on the regression results, the variables that were not statistically significant were removed using the backward screening method to obtain model 2.

#### 4.2.1. Individual Characteristics of Farmers

Farmers’ individual characteristics had a significant effect on STFFT adoption behavior. As shown in Table 2, the influence coefficient of farmers’ education level (IC3) is 0.644, indicating that this factor has a significant positive influence on farmers’ STFFT adoption behavior, and the empirical results are consistent with the expected direction of influence in the previous section. This indicates that, all other influences being equal, the higher the education level of farmers, the better their ability to understand and apply STFFT and the more motivated they are to adopt STFFT. The actual survey found that the less educated farmers were, the more they insisted on applying the traditional fertilizer application method for their current production activities and lacked accurate knowledge of STFFT. This is due to the fact that farmers are often in a state of information asymmetry during the technology diffusion process, and the influx of information at this time further reduces the uncertainty of production and enhances farmers’ knowledge of the relationship between agricultural production and ecological environment, making their judgments about the expected effects of technology adoption more accurate. In general, the more frequently farmers communicate with others and the faster the information flow, the more proactive they are in adopting STFFT.

#### 4.2.2. Family Characteristics and Business Characteristics of Farmers

The proportion of farming income to total family income (FC1) has a significant positive effect on farmers’ STFFT adoption behavior, and the empirical results are consistent with the previous hypothesis. All other influences being equal, the greater the reliance on agricultural production behavior for the group of farmers whose main source of income is farm income. This is because, unlike other groups of farmers with part-time nature, such groups of farmers have very limited access to family income, which makes them more closely tied to the land they cultivate. They invest more money, time and effort in the process of agricultural production, and perceive higher expected benefits of STFFT, which enhances the motivation to use this technology to a greater extent.

At the level of business characteristics, the larger the farmers’ planting scale (BC1), the easier they perceive the scale effect brought by STFFT and the more likely they are to achieve higher economic benefits from the process of technology application. Farmers’ farming scale has a direct impact on their economic returns, and the larger the farming scale, the more dependent farmers are on agricultural production and operation, the better they understand the link between technology application and field management and the more they pay attention to the types of technologies that can improve agricultural production. In contrast, small-scale farmer groups are constrained by conditions such as the scope of technology application and are less motivated to adopt STFFT.

#### 4.2.3. Cognitive Characteristics of Farmers

In addition to the influence of farmers’ individual, family and business characteristics, the “push” effect of their cognitive characteristics also influenced STFFT adoption behavior. As shown in Table 2, the higher the level of technology awareness (PC1), the more farmers know about STFFT and the easier it is to adopt the technology. The perceived ease of use of STFFT (PC4) had a significant positive effect on farmers’ technology adoption behavior. The less time, effort and money farmers psychologically perceive it takes to adopt STFFT, the more likely they are to adopt the technology. The actual survey found that STFFT requires farmers to master various fertilizer ratios and split applications, which is more complicated and tedious than traditional fertilizer application techniques, and farmers need to spend time and energy to learn and summarize, as well as invest a lot of labor to apply fertilizer. Farmers are more willing to adopt STFFT only when they think it is easier to master and operate and do not require more labor. For perceived usefulness (PC5), similarly, farmers will compare STFFT with existing fertilizer application technologies, and only when farmers truly recognize that the technology is superior to existing fertilizer application technologies will they be willing to adopt it, reflecting to a certain extent that fertilizer application technology adoption is a rational economic behavior.

#### 4.2.4. Characteristics of Farmers’ External Environment

Farmers’ STFFT adoption behavior was affected to different degrees by the stimulating effects of external environmental factors, such as government subsidies (EC2), technology training (EC4) and land transfer (EC5). Government subsidies can effectively weaken the influence on farmers’ technology adoption decisions due to risk preferences. Most farmers’ risk types belong to risk-averse type, and the intervention of subsidies can effectively reduce the risk of applicability and cost increase in STFFT, resulting in positive intervention effects. In addition, government subsidies enhance farmers’ livelihood resilience to a certain extent, giving them greater confidence to adopt new agricultural technologies and reducing their concerns about economic losses. Agricultural technology training, as a form of STFFT extension, can help farmers correctly understand the relationship between fertilizer input and crop yields, avoid wrong fertilizer input expectations and develop correct perceptions of fertilizer input effects. Large amounts of fertilizer application by farmers can increase the short-term output of arable land to a certain extent, but long-term overuse can result in consequences such as soil slumping and salinization. Such knowledge will encourage farmers to clarify the impact of STFFT on fertilizer inputs and ecological environment and change their unreasonable fertilizer application behavior. Additionally, farmers will transfer and integrate their idle land, which can promote the development of agricultural scale, which will increase the adoption demand for STFFT.

### 4.3. ISM Regression

According to the steps of the ISM analysis method, this study first determined the composition of the system. In this paper, S1, S2, S3, S4, S5, S6, S7, S8, S9 and S10 are used to denote the 10 significant variables that affect farmers’ STFFT adoption, i.e., education level, planting scale, technology training and land transfer. Through analysis, discussion and negotiation, the 10 elements that constitute the system and the logical relationships that exist were clarified to ensure the structure, hierarchy and openness of the system. The logical relationships of the main factors influencing farmers’ STFFT adoption behavior are shown in Figure 4.

Based on Figure 4, this study constructs the reachable matrix R that influences the adoption of STFFT by farmers, as follows:(5)$$R=\overset{}{\underset{}{}}{\begin{array}{c} S_{0}\\ S_{1}\\ S_{2}\\ S_{3}\\ S_{4}\\ S_{5}\\ S_{6}\\ S_{7}\\ S_{8}\\ S_{9}\\ S_{10} \end{array}}^{}{}_{}\left[\begin{array}{ccccccccccc} 1 & 0 & 0 & 0 & 0 & 0 & 0 & 0 & 0 & 0 & 0\\ 1 & 1 & 0 & 1 & 0 & 1 & 0 & 1 & 1 & 0 & 0\\ 1 & 0 & 1 & 0 & 1 & 0 & 0 & 1 & 1 & 0 & 0\\ 1 & 0 & 0 & 1 & 0 & 0 & 0 & 1 & 1 & 0 & 0\\ 1 & 0 & 0 & 0 & 1 & 0 & 0 & 1 & 1 & 0 & 0\\ 1 & 0 & 0 & 0 & 0 & 1 & 1 & 1 & 1 & 0 & 0\\ 1 & 0 & 0 & 0 & 0 & 0 & 1 & 0 & 0 & 0 & 0\\ 1 & 0 & 0 & 0 & 0 & 0 & 1 & 1 & 0 & 0 & 0\\ 1 & 0 & 0 & 0 & 0 & 0 & 1 & 0 & 1 & 0 & 0\\ 1 & 0 & 0 & 0 & 0 & 0 & 0 & 0 & 0 & 1 & 0\\ 1 & 0 & 0 & 0 & 0 & 0 & 0 & 0 & 0 & 0 & 1 \end{array}\right]$$

The structural model is to describe the correlation of the elements in the farmer’s adoption of the STFFT influence factor system using directed connection diagrams and eventually construct a system model that contains all the elements. In this study, the structural model will be constructed based on the reachable matrix R. As can be seen from Figure 4, the matrix elements in the rows of *S*_6_, *S*_9_ and *S*_10_ and the columns of *S*_0_ are all 1, which means that elements *S*_6_, *S*_9_ and *S*_10_ are all related to element *S*_0_, and, thus, the directed line segments from *S*_6_, *S*_9_ and *S*_10_ to *S*_0_ can be drawn. According to the above operation method, the structural model of the whole farmer adoption STFFT influence factor system can be drawn, as shown in Figure 5.

According to the ISM model shown in Figure 5, it can be intuitively seen that the factors influencing farmers’ STFFT adoption behavior are a typical multi-step structural model, with arrows pointing to indicate that lower-order (low-level) factors influence higher-order (high-level) factors. The first-level factors include land transfer, STFFT awareness and government subsidies, which directly influence farmers’ adoption of STFFT and belong to the surface-level direct factors that affect farmers’ adoption of STFFT. The second level factors include perceived ease of use of STFFT and perceived usefulness of STFFT, which directly influence farmers’ STFFT perceptions and are superficial indirect factors affecting farmers’ adoption of STFFT. The third level factors include technical training, information accessibility and the proportion of farming income to total family income, which directly influence the perceived ease of use of STFFT and the perceived usefulness of STFFT and belong to the middle level indirect factors influencing farmers’ adoption of STFFT. The fourth level factors include education level and farming scale, among which education level directly affects technical training and information acquisition ability, and farming scale directly affects the proportion of farming income to total family income, which are deep-rooted factors affecting farmers’ STFFT adoption. Taken together, farmers’ STFFT adoption behavior can be summarized into the following two major paths.

“Literacy–training–perception–stimulus–behavior” is the first path. Specifically, it is composed of the factors of educational level → technology training → perceived usefulness, perceived ease of use → land transfer, government subsidies, technology awareness → STFFT adoption behavior. In this path of influence, farmers’ education level and participation in technology training are the basic elements that influence farmers’ technology adoption behavior. Farmers’ education level is the basis for receiving new knowledge and is the original knowledge reserve. The higher is the basic knowledge of farmers, the stronger is their ability to learn new technologies when they receive technology training later on. The extension effect of STFFT has an important influence on adoption behavior, and this influence is continuous and diffuse. At the same time, farmers have a certain degree of herd mentality towards technology adoption, and when they face uncertainty in the future, they generally and directly show learning and imitation. Therefore, based on participation in training, farmers’ perceived usefulness and ease of use of STFFT will be greatly enhanced, and their probability level of STFFT adoption will be significantly increased with the intervention of external factors, such as government subsidies.

“Literacy–competence–perception–stimulus–behavior” is the second path. Specifically, it is composed of the factors of educational attainment → information acquisition ability → perceived usefulness, perceived ease of use → land transfer, government subsidies, and technology awareness → STFFT adoption behavior. In this influence path, farmers’ education level and information acquisition ability are the basic elements that affect farmers’ technology adoption behavior. Similar to the first path, the higher the education level of farmers, the better their knowledge absorption ability will be than other farmers with a lower education level, which is the basis of technology adoption. On this basis, the stronger the information acquisition ability of farmers, the more information they have access to technology, the more they will increase their understanding of STFFT and better grasp the process and principles of technology use. At this stage, the level of infrastructure construction in rural areas has been greatly improved, and the use of modern communication equipment has gradually become popular among farmers. The intervention of modern communication technology has enabled farmers to obtain information from a wider range of sources and make it easier to obtain information on new agricultural technologies. Through external support such as late government transfer payments and land transfer, farmers are more inclined to accept STFFT as a scientific fertilization technology.

The third path is “scale-income structure–perception–stimulus–behavior”. Specifically, it is composed of the factors of cultivation scale → proportion of cultivation income to total family income → perceived usefulness, perceived ease of use → land transfer, government subsidies, technology awareness → STFFT adoption behavior. In this path of influence, the scale of farmers’ production operations and family income structure are the underlying elements that influence farmers’ technology adoption behavior. The larger the scale of farmers’ engagement in agricultural production and operation, the larger the proportion of agricultural income in their overall family income profile. Such groups of farmers have a stronger economic base, are more dependent on agricultural production, are more sensitive to factor inputs in the agricultural production process and are more receptive to new agricultural technologies. With the weakening of China’s urban–rural dual structure and the improvement of basic transportation facilities, the mobility of agricultural labor between urban and rural areas accelerates and some groups of farmers gradually shift to part-time employment and their intra-family income structure changes. When agricultural income no longer occupies the main component of total family income, farmers’ motivation to engage in agricultural production activities is weakened, reducing their motivation to adopt new agricultural technologies. Therefore, groups of farmers with a certain scale of agricultural production and operation and whose main source of income are agriculture, they are more likely to adopt STFFT when promoted by external agents such as the government.

## 5. Discussion

### 5.1. Major Contributions Made in This Study

Existing studies on the use of STFFT among farmers have focused their perspectives more on influencing factors (external intervention elements, intra-individual characteristics), psychological decision-making processes, knowledge literacy and post-utility [14,16,68,69,70,71]. For example, Qi et al. conducted a study of 30 neighboring villages in Taojiang County using spatial measurement methods. They explored, in depth, the factors influencing farmers’ use of friendly fertilizer application technologies by combining family surveys, farmland quality surveys, remote sensing images and digital elevation models. The results showed that personal characteristics and topographic conditions influenced farmers’ decisions and that farmers with higher education levels were more likely to adopt new technologies [72]. Similar findings were obtained in this study that the higher the level of education received by farmers, the greater their knowledge base and the easier it is to adopt STFFT and apply it in practice. We used micro-survey data from 691 specialized apple farmers in Shandong and Shaanxi provinces to elucidate the effects of technical training and land operation size on STFFT adoption within the framework of the theory of planned behavior. The conclusions show that technical training experience significantly motivates farmers to use STFFT, and this motivating effect is more pronounced for the group of farmers with large-scale operations [73]. This study similarly verified the role of technology training and scale of operation in promoting farmers’ technology adoption behavior and obtained the pathways of action based on this. Similarly, Xue et al. used a logit linear regression model based on data from a sample of 700 farmers in the Loess Plateau and Bohai Bay regions to reconfirm that technology training has a differential impact on different size groups of farmers [74].

Compared with the extant literature, the present study may have the following marginal contributions. First, the study is more novel in terms of perspective. This study focuses on the adoption of STFFT among maize farmers within the black soil area of China, which is more focused compared to previous research areas, study subjects and crop types. The black soil area of China faces serious soil degradation and the growth process of the maize crop is closely related to the application of chemical fertilizers, based on which an in-depth investigation of STFFT-related situations can be more relevant. Second, the study is more systematic in content. This study constructs a logical structure of “external environmental stimulus–perceived characteristics–adoption behavior” under the framework of technology acceptance theory and proposes research hypotheses and test designs based on it. The external and internal factors that influence farmers’ decision-making behavior are fully considered and their internal mechanisms and mechanisms of action are further clarified. This will better facilitate the implementation of STFFT extension and improve production efficiency. Third, the study is more scientific in its approach. This paper adopts a combination of qualitative and quantitative analysis and integrates traditional linear regression models with path mining to make the conclusions more convincing. After using the logistic model to obtain the specific factors affecting farmers’ adoption of STFFT, the ISM model is used to further obtain the hierarchical structure and progressive direction among the factors, providing clearer paths and suggestions for improving technology adoption rates.

### 5.2. Deficiencies of the Study

This study also has the following shortcomings. First, considering the two important characteristics of large black soil distribution and STFFT first demonstration area, this study set the research area to 10 major grain-producing cities in Liaoning province. This makes the conclusions scientific, typical and reliable. However, it is a pity that other production areas in China have not been discussed due to the research conditions and other limitations. China is rich in land resources, with a wide variety of terrain and climate and diverse cultural backgrounds such as human history. The food production bases, mainly in the Yangtze River Delta region and the North China Plain, also have important research value. Therefore, it is still a question that needs further consideration as to what different degrees of regional location variability may bring to farmers’ STFFT adoption behavior. Therefore, in future studies, further adjustments can be made to the study area and the scope of the study can be expanded appropriately. Second, further improvements can be made in the use of research methods. This study uses a combination of qualitative and quantitative analysis, and although it has good results in the excavation of influencing factors and the analysis of influence paths, it still needs further in-depth excavation. Farmers’ STFFT adoption behavior belongs to an economic behavior, i.e., the behavior it occurs is rational. By rational, we mean that farmers’ STFFT adoption behavior is a rational behavior to maximize their own interests after fully and comprehensively weighing the benefits and costs under the given internal conditions and external environmental constraints. This behavior can be explained not only by theories of economics, but also by theories of psychology and behavioral science. Third, the main direction of this study is to investigate the behavioral decision of farmers to adopt STFFT. Although the factor of this decision process is the farmer, the involvement of other subjects in the formation of the decision cannot be ignored. The most directly related to farmers are the groups of technology promotion, which may be researchers, salespersons, service providers or production organizations such as cooperatives. Due to the limitation of the research sample, this study is limited to farmers, and the personal or organizational aspects involved in technology promotion are not discussed, which is a pity for this study. In the future, this study will attempt to explore in depth and in detail from a multi-subject perspective, which in turn will complement this study.

### 5.3. Future Research Prospects

During the future research period, the following aspects can be further explored in more depth. First, the research area should be further expanded. The environmental problems caused by chemical fertilizer abuse have caused damage to several soil species, and there are differences in farming habits between regions. Therefore, in the next study design, the sample acquisition area is not limited to Liaoning Province, but will be extended according to the specific research questions. The study area can be divided according to different climatic characteristics to form a controlled study. In addition, the time period of the study can be further extended. Although the cross-sectional data can clarify the influence mechanism among the factors, it is difficult to clarify the dynamic influence in the time series. Changes in farmers’ fertilizer application behavior, technology adoption behavior and soil physicochemical characteristics are not visible in the short term. In the next study, a dynamic tracer approach will be adopted to focus on the effect of farmers’ STFFT adoption on soil conservation over a long period of time. Second, an attempt was made to use different theories from different disciplines to explain farmers’ STFFT decision-making behavior. Farmers’ decision-making behavior is a long-term psychological process; therefore, in future research, we should try to switch our thinking and integrate the characteristics of multiple disciplines to interpret and explain based on different theories. In the selection of variables, different research methods can be studied. In addition to the quantification of indicators, the method of randomized trials can be applied to divide farmers into experimental and control groups to observe how farmers’ decision-making behaviors change under different intervention conditions. This research approach breaks down the barriers that make it difficult to observe counterfactual phenomena and allows for a more scientific analysis. Third, the perspective of the study is changed. Farmers are the main decision makers of STFFT adoption behavior, but in the market mechanism, farmers are also the recipients in the case of information asymmetry. At present, STFFT has not yet been adopted on a large scale, and the reasons for this do not only lie in the farmers’ group, but also in the reliability of the technology’s promoters and the efficiency of the cooperating institutions, which are matters that can hinder the adoption of the technology. Therefore, the research on STFFT adoption should not only focus on farmers, but also be further developed.

## 6. Conclusions

Based on a survey of 402 maize farmers in Liaoning Province, this paper constructed an analytical framework of “external environmental stimuli–perceived characteristics–decision making behavior” and explored the factors influencing farmers’ acceptance of STFFT through a logistic-ISM model. Three main conclusions are drawn below.

First, the surveyed farmers’ decision-making behaviors regarding STFFT were consistent with the analytical framework of technology acceptance theory and had significant profit propensity and risk aversion. Five categories of factors, namely, basic personal characteristics, family characteristics, business characteristics, cognitive characteristics and external environmental characteristics, have significant effects on farmers’ technology adoption behavior.

Second, in general, the level of education is a deep-rooted factor affecting farmers’ absorption of new agricultural technologies, and the more educated farmers are more likely to adopt STFFT. The depth of knowledge accumulation enhances farmers’ information acquisition ability, based on which the ratio of technical training and planting income is indirectly enhanced, which motivates farmers to generate technology adoption behavior.

Third, the influence of factors located under different strata on farmers’ STFFT adoption behavior is heterogeneous with multiple realization paths. Specifically, they can be divided into three intrinsic mechanisms, briefly summarized as two paths that unfold based on educational attainment, and one path that extends based on production scale.

Based on the above findings, this study can generate the following policy recommendations. First, accelerate rural land transfer. At present, China is still dominated by small-scale forms of agricultural operations, which, to a certain extent, limits the effectiveness of agricultural technologies. Therefore, the government can appropriately develop large-scale production, thus enabling the diffusion of agricultural technologies based on a more suitable starting point. Second, strengthen farmers’ perception of the usefulness of STFFT and actively achieve technology transformation and utilization. The government should increase the publicity effect of STFFT, so that farmers can understand the effect of the technology in improving crop yield and increasing income through the most intuitive way. This will enable farmers to gradually change their traditional fertilization concepts and establish a scientific fertilization knowledge system. Third, reduce the cost of technology adoption by farmers and strengthen STFFT training. The government should increase the support and financial subsidies for large-scale households and fertilizer producers. At the same time, technical extension departments should strengthen the organization of farmers to learn relevant knowledge, thus improving the STFFT adoption rate.

## Figures and Tables

**Figure 1 ijerph-19-15682-f001:**
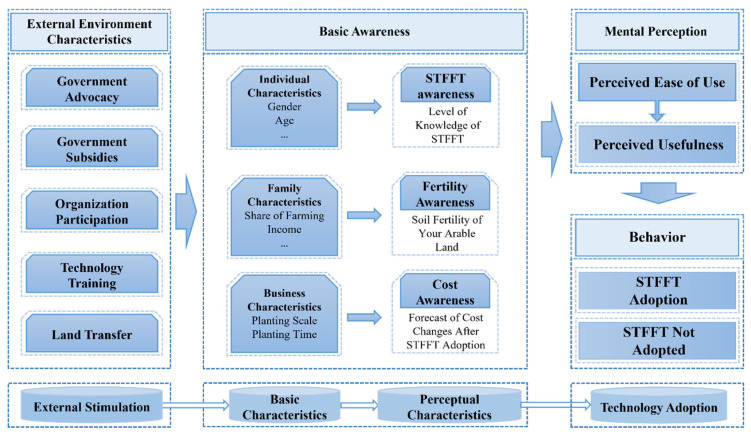
Theoretical analysis framework of “external environmental stimulus–perceived characteristics–adopted behavior”.

**Figure 2 ijerph-19-15682-f002:**
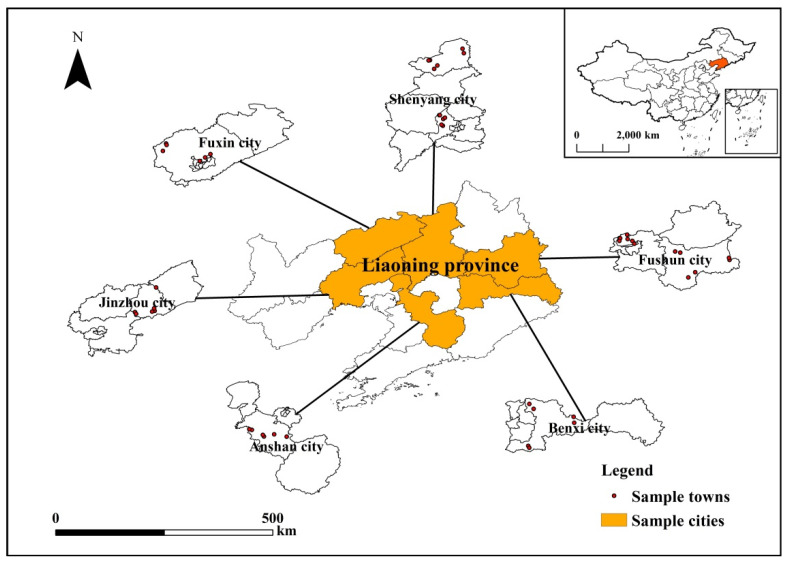
Map of the study area and spatial distribution of the sample villages.

**Figure 3 ijerph-19-15682-f003:**
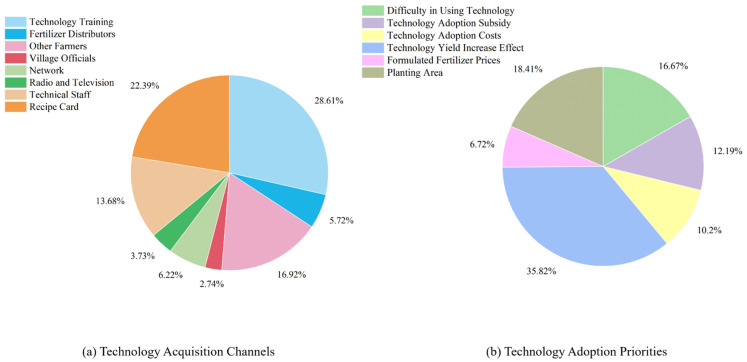
Map of background factors for farmers using STFFT.

**Figure 4 ijerph-19-15682-f004:**
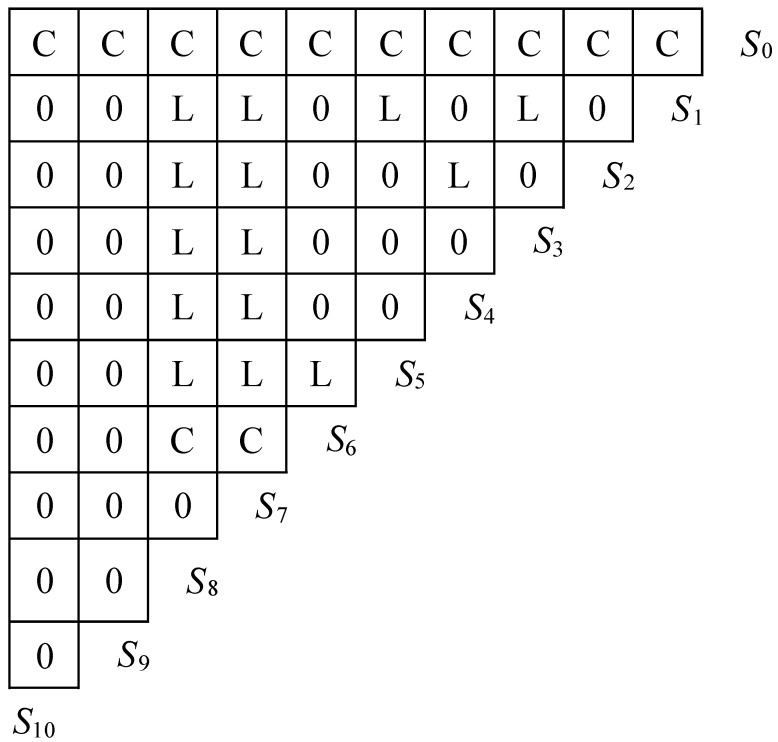
The logic relationship of elements. Where C indicates that the column element directly affects the row factor; L indicates that the row element directly affects the column factor; and 0 indicates that the row element has no relationship with the column factor.

**Figure 5 ijerph-19-15682-f005:**
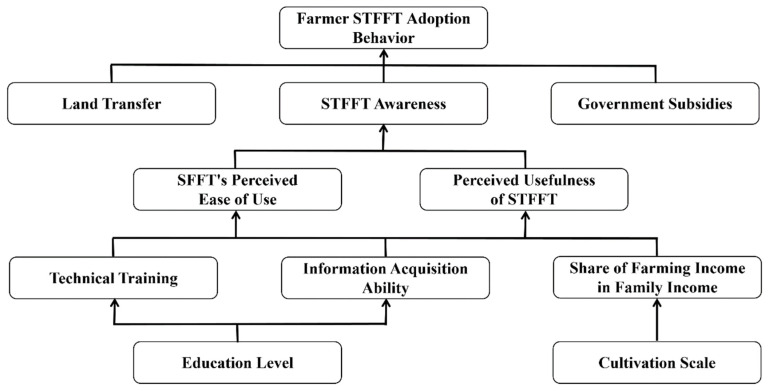
The interpretative structural model of factors.

**Table 1 ijerph-19-15682-t001:** The Interpretation and Statistical Nature of Model Variables.

Variables	Definition	Mean	Min	Max	S.D.	Predict
* **Dependent variable** *	Adopt STFFT or not? Not adopted = 0; adopted = 1	0.58	0	1	0.49	
* **Individual Characteristics** *						
IC1	Female = 0; male = 1	0.77	0	1	0.47	+
IC2	25 years old and younger = 1; 26 to 35 years old = 2; 36 to 45 years old = 3 46 to 55 years old = 4; 56 years old and above = 5	3.64	1	5	1.02	−
IC3	Elementary school and below = 1; middle school = 2; high school or junior college = 3; college and above = 4	1.58	1	4	0.81	+
IC4	Very weak = 1; weak = 2; fair = 3; strong = 4; very strong = 5	3.48	1	5	0.93	+
* **Family Characteristics** *						
FC1	60% and below = 1; 61% to 70% = 2; 71% to 80% = 3; 81% to 90% = 4; 91% and above = 5	3.47	1	5	0.98	+
FC2	20% and below = 1; 21% to 40% = 2; 41% to 60% = 3; 61% to 80% = 4; 81% and above = 5	2.27	1	5	1.37	−
* **Businesses Characteristics** *						
BC1	5 acres and below = 1; 5 to 10 acres = 2; 11 to 15 acres = 3; 16 to 20 acres = 4; 21 acres and above = 5	3.32	1	5	1.07	+
BC2	5 years and below = 1; 5 to 10 years = 2; 11 to 15 years = 3; 16 to 20 years = 4; 21 years and above = 5	3.41	1	5	1.11	+
* **Perception Characteristics** *						
PC1	Awareness of STFFT: very unaware = 1; relatively unaware = 2; average = 3; relatively aware = 4; very aware = 5	3.64	1	5	1.12	+
PC2	Perceived fertility of arable land: very bad = 1; relatively bad = 2; fair = 3; relatively good = 4; very good = 5	2.79	1	5	1.04	−
PC3	Adopting STFFT would reduce costs: strongly disagree = 1; relatively disagree = 2; average = 3; relatively agree = 4; strongly agree = 5	3.39	1	5	0.99	+
PC4	STFFT can be grasped more easily: strongly disagree = 1; relatively disagree = 2; average = 3; relatively agree = 4; strongly agree = 5	3.21	1	5	1.08	+
PC5	STFFT has a greater effect on production: strongly disagree = 1; relatively disagree = 2; average = 3; relatively agree = 4; strongly agree = 5	3.36	1	5	1.04	+
* **Environmental Characteristics** *						
EC1	Government advocacy STFFT: no = 0; yes = 1	0.68	0	1	0.47	+
EC2	Government subsidy STFFT: no = 0; yes = 1	0.65	0	1	0.48	+
EC3	Has joined the industrial cooperative organization: no = 0; yes = 1	0.65	0	1	0.48	+
EC4	Whether trained in agricultural technology: no = 0; yes = 1	0.53	0	1	0.50	+
EC5	Whether to carry out land transfer: no = 0; yes = 1	0.41	0	1	0.49	+

**Table 2 ijerph-19-15682-t002:** The result of logistic model regression analysis.

Variables	Model 1				Model 2			
	Coefficient	Std. Err.	Marginal Effects	Std. Err.	Coefficient	Std. Err.	Marginal Effects	Std. Err.
IC1	0.349	0.388	0.031	0.035	-	-	-	-
IC2	− 0.004	0.163	−0.001	0.015	-	-	-	-
IC3	0.703 ***	0.274	0.063 ***	0.024	0.644 **	0.259	0.060 **	0.023
IC4	0.589 **	0.232	0.053 ***	0.020	0.591 ***	0.224	0.055 ***	0.020
FC1	0.603 **	0.236	0.054 ***	0.021	0.695 ***	0.219	0.065 ***	0.019
FC2	−0.064	0.165	−0.006	0.015	-	-	-	-
BC1	0.961 ***	0.219	0.087 ***	0.018	1.048 ***	0.209	0.098 ***	0.017
BC2	0.179	0.196	0.016	0.018	-	-	-	-
PC1	0.416 **	0.179	0.037 **	0.016	0.469 ***	0.162	0.044 ***	0.014
PC2	0.052	0.163	0.005	0.015	-	-	-	-
PC3	0.267	0.219	0.024	0.020	-	-	-	-
PC4	0.765 ***	0.178	0.069 ***	0.015	0.744 ***	0.169	0.069 ***	0.014
PC5	0.655 ***	0.200	0.059 ***	0.017	0.674 ***	0.199	0.063 ***	0.018
EC1	0.450	0.393	0.040	0.035	-	-	-	-
EC2	0.930 **	0.472	0.084 **	0.042	0.677 **	0.430	0.063 **	0.040
EC3	0.422	0.363	0.038	0.033	-	-	-	-
EC4	0.921 *	0.497	0.083 *	0.044	1.251 ***	0.338	0.117 ***	0.030
EC5	0.665 *	0.355	0.060 *	0.031	0.704 **	0.336	0.066 **	0.031
Constant	−17.595 ***	2.041			− 15.997 ***	1.716		
Number of total observations	402		402		402		402	
LR chi^2^	312.20				304.72			
Log likelihood	−117.430				−121.171			
Prob > chi^2^	0.000				0.000			
Pesudo R^2^	0.571				0.557			

Note: Standard errors are in parentheses; *** significant at *p* < 0.01, ** significant at *p* < 0.05, and * significant at *p* < 0.1.

## Data Availability

The data presented in this study are available on request from the corresponding author. The data are not publicly available due to privacy restrictions.

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
