# Peer review of "What Influences Farmers’ Adoption of Soil Testing and Formulated Fertilization Technology in Black Soil Areas? An Empirical Analysis Based on Logistic-ISM Model"

_ijerph, 2022, doi:10.3390/ijerph192315682_

Round 1
Reviewer 1 Report
The paper applies theoretical framework to investigate the decision-making and the dynamic influence mechanism of farmers' adoption behavior of STFFT. The paper presents interesting empyrical study results for China. The paper is well-written and has well-developed structure. The paper needs revision to provide more clarity on approach selected. What are the main strengths and limits of proposed approach in comparison with other studies in this field. There are many abbreviations in this paper therefore I would suggest to include the list of abbreviations in the paper to make reading of this paper easier. The conclusions need to be rewritten and better structured. The novelty and input of this paper need to be highlighted. I also was able to find more new sources on the subject as well.
Author Response
Dear Editor:
Thank you for giving us a chance to improve the manuscript, entitled “What Influences Farmers' Adoption of Soil Testing and Formulated Fertilization Technology in Black Soil Areas? An Empirical Analysis Based on Logistic-ISM Model” (ID: ijerph-2007390). We appreciate the constructive comments from anonymous reviewers, which are very helpful for us revising and improving our paper. We have studied the comments carefully and have made necessary corrections accordingly. We believe the manuscript has significantly improved.
To better show what has been changed, we enclose the manuscript in "Track Changes" mode. In addition, we summarize the point-by-point response as below. Note that the Lines numbers mentioned in the following responses are according to the revised manuscript. And our responses are marked in Blue.
Detailed responses to the reviewer’s comments:
Point 1: The paper needs revision to provide more clarity on approach selected. What are the main strengths and limits of proposed approach in comparison with other studies in this field.
Response 1: We are grateful for such a valuable suggestion. We have deleted the sentence "This combination breaks through the limitations of using quantitative or qualitative analysis alone, and after obtaining the direction and extent of the factors influencing farmers' adoption of STFFT through regression analysis, the interrelationships and hierarchical structure among the factors are further analyzed through ISM analysis" between lines 333-337 in the manuscript and inserted the following sentence between lines 416-428:
“Such a combination breaks through the limitations of using quantitative or qualitative analysis alone. After using regression analysis to obtain the direction and extent of the factors that influence farmers' adoption of STFFT, the interrelationships and hierarchical structure between the factors can be further obtained through ISM analysis. Compared with existing studies using models such as simple linear and structural equations [14,62], this study breaks through the limitations of quantitative analysis and combines qualitative with it. This allows for a clearer formation of a stepwise progression of relationships and a clarification of the intrinsic connections between elements. Although this combination of Logistic-ISM has the advantage of analytical channel expansion, it has some limitations in the duality of dependent variables and the capacity of the factor set. When the number of elements is too large, the output of such complex matrix algorithms and graphs makes the computing time grow exponentially.”
Point 2: There are many abbreviations in this paper therefore I would suggest to include the list of abbreviations in the paper to make reading of this paper easier.
Response 2: Great idea! We appreciate this valuable comment from you. Thanks! We have included a list of abbreviations at the end of the manuscript text.
Point 3: The conclusions need to be rewritten and better structured.
Response 3: Thanks! For improving the content of the conclusion section, we have eliminated redundant text and restructured the exposition of the conclusion section. The sentence "This study constructs an analytical framework of…Third, these 10 salience factors are both independent and interrelated with each other, and together they form a structural system of factors influencing farmers' adoption of STFFT" between lines 817-840 in the manuscript has been removed and the following sentence has been inserted between lines 986-1011.
“Based on a survey of 402 maize farmers in Liaoning Province, this paper constructed an analytical framework of "external environmental stimuli-perceived characteristics-decision making behavior" and explored the factors influencing farmers' acceptance of STFFT through a logistic-ISM model. Three main conclusions are drawn below.
First, the surveyed farmers' decision-making behaviors regarding STFFT were consistent with the analytical framework of technology acceptance theory and had significant profit propensity and risk aversion. Five categories of factors, namely, basic personal characteristics, family characteristics, business characteristics, cognitive characteristics, and external environmental characteristics, have significant effects on farmers' technology adoption behavior.
Second, in general, the level of education is a deep root factor affecting farmers' absorption of new agricultural technologies, and the more educated farmers are more likely to adopt STFFT. the depth of knowledge accumulation enhances farmers' information acquisition ability, based on which the ratio of technical training and planting income is indirectly enhanced, which motivates farmers to generate technology adoption behavior.
Third, the influence of factors located under different strata on farmers' STFFT adoption behavior is heterogeneous with multiple realization paths. Specifically, they can be divided into three intrinsic mechanisms, briefly summarized as two paths that unfold based on educational attainment, and one path that extends based on production scale.”
Point 4: The novelty and input of this paper need to be highlighted. I also was able to find more new sources on the subject as well.
Response 4: Thanks for your precious comments. We have removed the sentence "Compared with the extant literature, the present study may have the following marginal contributions…which can provide new research ideas for other related studies on agricultural technology adoption." from the manuscript between lines 868-887.
“Compared with the extant literature, the present study may have the following marginal contributions. First, the study is more novel in terms of perspective. This study focuses on the adoption of STFFT among maize farmers within the black soil area of China, which is more focused compared to previous research areas, study subjects, and crop types. The black soil area of China faces serious soil degradation, and the growth process of maize crop is closely related to the application of chemical fertilizers, based on which an in-depth investigation of STFFT related situations can be more relevant. Second, the study is more systematic in content. This study constructs a logical structure of "external environmental stimulus - perceived characteristics - adoption behavior" under the framework of technology acceptance theory, and proposes research hypotheses and test designs based on it. The external and internal factors that influence farmers' decision-making behavior are fully considered, and their internal mechanisms and mechanisms of action are further clarified. This will better facilitate the implementation of STFFT extension and improve production efficiency. Third, the study is more scientific in its approach. This paper adopts a combination of qualitative and quantitative analysis, and integrates traditional linear regression models with path mining to make the conclusions more convincing. After using the logistic model to obtain the specific factors affecting farmers' adoption of STFFT, the ISM model is used to further obtain the hierarchical structure and progressive direction among the factors, providing clearer paths and suggestions for improving technology adoption rates.”
Correspondingly, we have replaced items 68-74 in the reference list with the updated literature. Once again, we would like to express our sincere gratitude for your suggestions.

Reviewer 2 Report
Reviewed manuscript “What Influences Farmers' Adoption of Soil Testing and Formulated Fertilization Technology in Black Soil Areas? An Empirical Analysis Based on Logistic-ISM Model” is an original and interesting study. Authors comprehensively demonstrated the Soil Testing and Formulated Fertilization Technology in Black Soil Areas. I would suggest minor revision.
Following are some suggestions for further improvements:
First few lines of the abstract should be about the importance of study.
Lines 46-50: Consider revising the lines.
Lines 67-69: Consider revising the lines.
Lines 788-790: Consider revising the lines.
Introduction and Discussion section needs to further strengthen by latest studies on the subject.
At some places in the text, there are grammatical mistakes that need to be corrected by some native English colleague.
To further strengthen introduction, following latest studies etc. are suggested to cite for the importance Soil Testing and Formulated Fertilization Technology in Black Soil Areas.
Author Response
Dear Editor:
Thank you for giving us a chance to improve the manuscript, entitled “What Influences Farmers' Adoption of Soil Testing and Formulated Fertilization Technology in Black Soil Areas? An Empirical Analysis Based on Logistic-ISM Model” (ID: ijerph-2007390). We appreciate the constructive comments from anonymous reviewers, which are very helpful for us revising and improving our paper. We have studied the comments carefully and have made necessary corrections accordingly. We believe the manuscript has significantly improved.
To better show what has been changed, we enclose the manuscript in "Track Changes" mode. In addition, we summarize the point-by-point response as below. Note that the Lines numbers mentioned in the following responses are according to the revised manuscript. And our responses are marked in Blue.
Detailed responses to the reviewer’s comments:
Point 1: First few lines of the abstract should be about the importance of study.
Response 1: We would like to appreciate your valuable comments. To add emphasis to the importance of this study, we have deleted the sentence "Clarifying the main factors affecting farmers' adoption of soil testing and formulated fertilization technology (STFFT) can further improve the technology adoption rate and fertilizer utilization efficiency, promote standardized agricultural production, and maintain a healthy and stable soil ecology in black soil areas" between lines 9-12 in the manuscript and inserted the following sentence between lines 9-16:
“Along with the increasing prominence of environmental risks such as soil surface source pollution and declining quality grade of arable land, the issues of how to address irrational fertilizer application and enhance the safety of agricultural products have attracted widespread attention. In this context, clarifying the main factors affecting farmers' use of soil testing and formulated fertilization technology (STFFT) can further improve the technology adoption rate and fertilizer utilization efficiency, promote standardized agricultural production, and maintain the health and stability of soil ecology in black soil areas. This is of great significance to the construction of green agriculture, national dietary health and national food security.”
Point 2: Lines 46-50: Consider revising the lines.
Response 2: Thanks! We have deleted the sentence "This irrational fertilizer application behavior can, to a certain extent, significantly observe the yield increase of agricultural products in the short term, but the long-term application of this method can cause consequences such as soil consolidation, erosion, and salinization, which further deteriorate the soil environment, destroy ecological stability, and expand agricultural surface source pollution" between lines 46-50 in the manuscript and inserted the following sentence between lines 64and 68:
“This unreasonable fertilizer input behavior can make farmers obviously observe the increase of agricultural products yield in the short term, but the long-term application of this method will cause a series of consequences such as soil slabbing, erosion and salinization, which will further deteriorate the soil environment, destroy the stability of the ecosystem and expand the agricultural surface pollution.”
Point 3: Lines 67-69: Consider revising the lines.
Response 3: Thanks for your precious comments, sincerely. We have deleted the sentence "To further improve the soil quality level of farmland, improve the ecological environment, promote agricultural carbon peaking and carbon neutrality, and enhance the efficiency of fertilizer utilization, STFFT has been gradually promoted and accepted [13-15]" between lines 67-69 in the manuscript and inserted the following sentence between lines 96-99:
“To further enhance the efficiency of fertilizer utilization and improve the soil quality level and ecological environment of farmland, STFFT is gradually promoted and accepted [13,14]. The proliferation of this technology has promoted agricultural carbon peaking and carbon neutrality [15].”
Point 4: Lines 788-790: Consider revising the lines.
Response 4: Thank you for your valuable suggestions. We have deleted the sentence "First, the design and selection of the research area will be further enriched. In the next study, the research area will not be limited to Liaoning Province or northeastern China. The environmental problems caused by the irrational application of chemical fertilizers are not only harmful to the black soil, so the research area will be further expanded" between lines 788-790 in the manuscript and inserted the following sentence between lines 952 and 956:
“First, the research area should be further expanded. The environmental problems caused by chemical fertilizer abuse have caused damage to several soil species, and there are differences in farming habits between regions. Therefore, in the next study design, the sample acquisition area is not limited to Liaoning Province, but will be extended according to the specific research questions.”
Point 5: Introduction and Discussion section needs to further strengthen by latest studies on the subject.
Response 5: Thanks for your valuable suggestions. We have further enriched the introduction section and the discussion section by studying and sorting out the latest literature.
First, we have carefully studied and discussed the latest literature, and amended the literature citations in the introduction section as follows.
We have deleted the fourth reference in the manuscript and replaced it with "Chen, X.; Ma, L.; Ma, W.; Wu, Z.; Cui, Z.; Hou, Y.; Zhang, F. What has caused the use of fertilizers to skyrocket in China?. Nutrient cycling in agroecosystems 2018, 110, 241-255. https://doi.org/10.1007/s10705-017-9895-1"
We have deleted the fifth reference in the manuscript and replaced it with "Wu, Y.; Xi, X.; Tang, X.; Luo, D.; Gu, B.; Lam, S.K.; Vitousek, P.M.; Chen, D. Policy distortions, farm size, and the overuse of agricultural chemicals in China. Proceedings of the National Academy of Sciences 2018, 115, 7010-7015. https://doi.org/10.1073/pnas.1806645115"
We have deleted the sixth reference in the manuscript and replaced it with "Chen, X.; Yu, W.; Cai, Y.; Zhang, S.; Muneer, M.A.; Zhu, Q.; Xu, D.; Ma, C.; Yan, X.; Li, Y.; Huang, S.; Wu, L.; Zhou, S.; Zhang, F. How to identify and adopt cleaner strategies to improve the continuous acidification in orchard soils?. Journal of Cleaner Production 2022, 330, 129826. https://doi.org/10.1016/j.jclepro.2021.129826"
We have deleted the seventh reference in the manuscript and replaced it with "Chen, X.; Yan, X.; Wang, M.; Cai, Y.; Weng, X.; Su, D.; Guo, J.; Wang, W.; Hou, Y.; Ye, D.; Zhang, S.; Liu, D.; Tong, L.; Xu, X.; Zhou, S.; Wu, L.; Zhang, F. Long-term excessive phosphorus fertilization alters soil phosphorus fractions in the acidic soil of pomelo orchards. Soil and Tillage Research 2022, 215, 105214. https://doi.org/10.1016/j.still.2021.105214"
We have deleted the ninth reference in the manuscript and replaced it with "Raza, S.; Miao, N.; Wang, P.; Ju, X.; Chen, Z.; Zhou, J.; Kuzyakov, Y. Dramatic loss of inorganic carbon by nitrogen‐induced soil acidification in Chinese croplands. Global change biology 2020, 26, 3738-3751. https://doi.org/10.1111/gcb.15101"
We have deleted the tenth reference in the manuscript and replaced it with "Ren, C.; Jin, S.; Wu, Y.; Zhang, B.; Kanter, D.; Wu, B.; Xi, X.; Xin, Z.; De, C.; Jian, X.; Gu, B. Fertilizer overuse in Chinese smallholders due to lack of fixed inputs. Journal of Environmental Management 2021, 293, 112913. https://doi.org/10.1016/j.jenvman.2021.112913"
We have deleted the eleventh reference in the manuscript and replaced it with "Zhu, Q.; de Vries, W.; Liu, X.; Hao, T.; Zeng, M.; Shen, J.; Zhang, F. Enhanced acidification in Chinese croplands as derived from element budgets in the period 1980–2010. Science of the Total Environment 2018, 618, 1497-1505. https://doi.org/10.1016/j.scitotenv.2017.09.289"
Second, after our careful literature review and discussion, the discussion section has been further improved. The sentence "Existing studies on farmers' technology adoption have focused their perspectives more on broad categories of agricultural …For example, …factors influencing pesticide technology preferences and finally verified the order of farmers' preferences and the main influencing factors under different technologies [74]" between lines 720-734 in the manuscript has been deleted and the following sentence has been inserted between lines 832-858:
“Existing studies on the use of STFFT among farmers have focused their perspectives more on influencing factors (external intervention elements, intra-individual characteristics), psychological decision-making processes, knowledge literacy and post-utility [14,16,68-71]. For example, Qi et al. conducted a study of 30 neighboring villages in Taojiang County using spatial measurement methods. They explored in depth the factors influencing farmers' use of friendly fertilizer application technologies by combining family surveys, farmland quality surveys, remote sensing images, and digital elevation models. The results showed that personal characteristics and topographic conditions influenced farmers' decisions, and that farmers with higher education levels were more likely to adopt new technologies [72]. Similar findings were obtained in this study that the higher the level of education received by farmers, the greater their knowledge base and the easier it is to adopt STFFT and apply it in practice. Wen used microsurvey data from 691 specialized apple farmers in Shandong and Shaanxi provinces to elucidate the effects of technical training and land operation size on STFFT adoption within the framework of the theory of planned behavior. The conclusions show that technical training experience significantly motivates farmers to use STFFT, and this motivating effect is more pronounced for the group of farmers with large-scale operations [73]. This study similarly verified the role of technology training and scale of operation in promoting farmers' technology adoption behavior and obtained the pathways of action based on this. Similarly, Xue et al. used a logit linear regression model based on data from a sample of 700 farmers in the Loess Plateau and Bohai Bay regions to reconfirm that technology training has a differential impact on different size groups of farmers [74].”
Correspondingly, we have replaced items 68-74 in the reference list with the updated literature. Once again, we would like to express our sincere gratitude for your suggestions.
Point 6: At some places in the text, there are grammatical mistakes that need to be corrected by some native English colleague.
Response 6: Thanks! We have sought help from native English colleagues regarding grammatical issues and have made corrections in the manuscript.
Point 7: To further strengthen introduction, following latest studies etc. Are suggested to cite for the importance Soil Testing and Formulated Fertilization Technology in Black Soil Areas.
Response 7: Thanks for your precious suggestions. We have deleted the sentence "As the most important production factor input in the process of food cultivation, land resources are one of the key factors to ensure the structural integrity of the agricultural ecosystem, and the conservation and sustainable use of land resources is an important foundation for national economic development and social harmony and stability [9-12]" and inserted the following sentence between lines 81-86.
“As the most important production factor in the process of food cultivation, land resources are one of the key factors to guarantee the structural integrity of the agroecological system, and their conservation and sustainable use is an important foundation for national economic development and social harmony and stability [9-11]. As one of the rare soil resources, black soil is an object of urgent attention and protection [12].”
At the same time, the 12th reference in the original manuscript was updated to “Zhou, M.; Liu, C.; Wang, J.; Meng, Q.; Yuan, Y.; Ma, X.; Liu, X.; Zhu, Y.; Ding, G.; Zhang, J.; Zeng, X.; Du, W. Soil aggregates stability and storage of soil organic carbon respond to cropping systems on Black Soils of Northeast China. Scientific Reports 2020, 10, 1-13. https://doi.org/10.1038/s41598-019-57193-1”
Special thanks to you for your good comments. We tried our best to improve the manuscript and made some changes in the manuscript. It is hoped that the correction will meet with approval.
Once again, thank you very much for your comments and suggestions.

Reviewer 3 Report
Dear Editor
I am sharing my review of the manuscript “ijerph-2007390 What Influences Farmers' Adoption of Soil Testing and Formulated Fertilization Technology in Black Soil Areas?”. The manuscript shows results from research in a China Province. The manuscript is regional and the authors write in discussion “1-This study also has the following shortcomings. First, the research area of this study is designed as 8 major grain-producing cities in Liaoning Province in northeastern China, which is still relatively limited in scope despite its typicality and scientificity. 2- The crop growth cycle in the southern part of China is different from that in the northern part of the country, and the water content and organic matter content of the soil are not exactly the same”. Comments and doubts in manuscript.
Best regards!

Author Response
Dear Editor:
Thank you for giving us a chance to improve the manuscript, entitled “What Influences Farmers' Adoption of Soil Testing and Formulated Fertilization Technology in Black Soil Areas? An Empirical Analysis Based on Logistic-ISM Model” (ID: ijerph-2007390). We appreciate the constructive comments from anonymous reviewers, which are very helpful for us revising and improving our paper. We have studied the comments carefully and have made necessary corrections accordingly. We believe the manuscript has significantly improved.
To better show what has been changed, we enclose the manuscript in "Track Changes" mode. In addition, we summarize the point-by-point response as below. Note that the Lines numbers mentioned in the following responses are according to the revised manuscript. And our responses are marked in Blue.
Detailed responses to the reviewer’s comments:
Point: The manuscript is regional and the authors write in discussion “1-This study also has the following shortcomings. First, the research area of this study is designed as 8 major grain-producing cities in Liaoning Province in northeastern China, which is still relatively limited in scope despite its typicality and scientificity. 2- The crop growth cycle in the southern part of China is different from that in the northern part of the country, and the water content and organic matter content of the soil are not exactly the same”. Comments and doubts in manuscript.
Response: Thank you for your precious comments, sincerely. There are two reasons for choosing Liaoning province as the research area in this study. First, Liaoning Province is an important black soil distribution area in China, and its area reaches 54,187.24 km2, accounting for 12.15% of the total area of the northeast black soil region. During 2015-2016, Liaoning Province arranged funding CNY29,270,000 in the northeast black soil conservation and utilization pilot project. An in-depth study on the adoption of soil formula fertilization technology in this region will help to better contain soil quality and optimize soil environment. Second, since 2005, Liaoning Province has promoted soil formula fertilization technology more than 460 million mu times, and established a total of 100,000 mu of various technology demonstration areas. During this period, more than 32,000 soil samples were collected, more than 128,000 sets of effective experimental data were obtained, and more than 275,000 copies of soil formula fertilization recommendation cards were issued (Liaoning Statistical Yearbook 2017). Thus, it can be seen that soil formula fertilization technology has experienced a period of development in Liaoning Province, and it is more appropriate to explore farmers' willingness to adopt it on this basis.
In summary, in order to better express the shortcomings of this study while ensuring the typicality of the sample, we have deleted the sentence "First, the research area of this study is designed as 8 major grain-producing cities in Liaoning Province in northeastern China, …… Therefore, the extent …… expanded appropriately" between lines 756-767 in the manuscript and inserted the following sentence between lines 909 and 920:
“First, considering the two important characteristics of large black soil distribution and STFFT first demonstration area, this study set the research area to 10 major grain-producing cities in Liaoning province. This makes the conclusions scientific, typical and reliable. However, it is a pity that other production areas in China have not been discussed due to the research conditions and other limitations. China is rich in land resources, with a wide variety of terrain and climate, and diverse cultural back-grounds such as human history. The food production bases, mainly in the Yangtze River Delta region and the North China Plain, also have important research value. Therefore, it is still a question that needs further consideration as to what different degrees of regional location variability may bring to farmers' STFFT adoption behavior. Therefore, in future studies, further adjustments can be made to the study area and the scope of the study can be expanded appropriately.”

Reviewer 4 Report
Although there are many studies on farmer perception, farmer cognition and farmer trust, etc. I haven't seen such a good article in a long time. The article is well organized, and the farmers' analysis is also good.
Introduction:I don't think this research is necessary to rub off on the COVID-19 hotspot because there is no direct relationship. Will farmers' perceptions change because of COVID-19?
I think some of your references are not very appropriate in the Introduction section, suggest citing DOI: 10.1016/j.jclepro.2021.129826; 10.1016/j.still.2021.105214; and 10.1007/s10705-017-9895-1
https://doi.org/10.1111/gcb.15101, https://doi.org/10.1016/j.scitotenv.2017.09.289, https://doi.org/10.1016/j.jenvman.2021.112913, https://doi.org/10.1073/pnas.1806645115.
. These articles can respectively illustrate the problem of excessive fertilization and environmental impact of different crops in China.
The vertical fonts in Figure 1 are recommended to be arranged horizontally at the top of the corresponding box, which is convenient for readers to read.
Be careful, the map of China in Figure 2 lacks a scale bar. The expansion of the area does not require two lines, one line is OK.
It is detailed that the questionnaire survey sample is well representative, which is very important for social science research, otherwise the results will be biased. But what you said is a bit too detailed, the same sentence can also be changed. For example, multiple occurrences of ‘In order to’.
It may be better to change Table 2 to a pie chart or a percentage histogram to enrich the look and feel of the article.
The discussion is not deep enough, and the discussion is to further enrich, demonstrate or extend the results on the basis of summarizing the literature. You are more self-analyzing than citing.
The conclusion is too long, too long, too long, shorten it to less than 400 words. This is not a conclusion, more like a review article.
Author Response
Dear Editor:
Thank you for giving us a chance to improve the manuscript, entitled “What Influences Farmers' Adoption of Soil Testing and Formulated Fertilization Technology in Black Soil Areas? An Empirical Analysis Based on Logistic-ISM Model” (ID: ijerph-2007390). We appreciate the constructive comments from anonymous reviewers, which are very helpful for us revising and improving our paper. We have studied the comments carefully and have made necessary corrections accordingly. We believe the manuscript has significantly improved.
To better show what has been changed, we enclose the manuscript in "Track Changes" mode. In addition, we summarize the point-by-point response as below. Note that the Lines numbers mentioned in the following responses are according to the revised manuscript. And our responses are marked in Blue.
Detailed responses to the reviewer’s comments:
Point 1: Introduction:I don't think this research is necessary to rub off on the COVID-19 hotspot because there is no direct relationship. Will farmers' perceptions change because of COVID-19?
Response 1: Thank you for this valuable suggestion. We have deleted the sentence "Under the influence of unstable factors such as global COVID-19 outbreaks, frequent extreme weather events and complex changes in the supply chain of agricultural products, how to increase food production and improve the efficiency and quality of agricultural production is a global and important issue to ensure national and regional food security and social stability" between lines 36-40 in the manuscript and inserted the following sentence between lines 44-47:
“Under the influence of unstable factors such as frequent global extreme weather events and complex changes in the supply chain of agricultural products, how to increase food yield and improve agricultural production efficiency is an important global issue to ensure regional food security and social stability”
Point 2: I think some of your references are not very appropriate in the Introduction section, suggest citing DOI:
10.1016/j.jclepro.2021.129826; 10.1016/j.still.2021.105214; and
10.1007/s10705-017-9895-1
https://doi.org/10.1111/gcb.15101,
https://doi.org/10.1016/j.scitotenv.2017.09.289,
https://doi.org/10.1016/j.jenvman.2021.112913,
https://doi.org/10.1073/pnas.1806645115.
These articles can respectively illustrate the problem of excessive fertilization and environmental impact of different crops in China.
Response 2: Thanks! Your valuable suggestions and recommended references have been carefully studied and discussed, and the following corrections have been made to the literature citations in the introduction section.
We have deleted the fourth reference in the manuscript and replaced it with "Chen, X.; Ma, L.; Ma, W.; Wu, Z.; Cui, Z.; Hou, Y.; Zhang, F. What has caused the use of fertilizers to skyrocket in China?. Nutrient cycling in agroecosystems 2018, 110, 241-255. https://doi.org/10.1007/s10705-017-9895-1"
We have deleted the fifth reference in the manuscript and replaced it with "Wu, Y.; Xi, X.; Tang, X.; Luo, D.; Gu, B.; Lam, S.K.; Vitousek, P.M.; Chen, D. Policy distortions, farm size, and the overuse of agricultural chemicals in China. Proceedings of the National Academy of Sciences 2018, 115, 7010-7015. https://doi.org/10.1073/pnas.1806645115"
We have deleted the sixth reference in the manuscript and replaced it with "Chen, X.; Yu, W.; Cai, Y.; Zhang, S.; Muneer, M.A.; Zhu, Q.; Xu, D.; Ma, C.; Yan, X.; Li, Y.; Huang, S.; Wu, L.; Zhou, S.; Zhang, F. How to identify and adopt cleaner strategies to improve the continuous acidification in orchard soils?. Journal of Cleaner Production 2022, 330, 129826. https://doi.org/10.1016/j.jclepro.2021.129826"
We have deleted the seventh reference in the manuscript and replaced it with "Chen, X.; Yan, X.; Wang, M.; Cai, Y.; Weng, X.; Su, D.; Guo, J.; Wang, W.; Hou, Y.; Ye, D.; Zhang, S.; Liu, D.; Tong, L.; Xu, X.; Zhou, S.; Wu, L.; Zhang, F. Long-term excessive phosphorus fertilization alters soil phosphorus fractions in the acidic soil of pomelo orchards. Soil and Tillage Research 2022, 215, 105214. https://doi.org/10.1016/j.still.2021.105214"
We have deleted the ninth reference in the manuscript and replaced it with "Raza, S.; Miao, N.; Wang, P.; Ju, X.; Chen, Z.; Zhou, J.; Kuzyakov, Y. Dramatic loss of inorganic carbon by nitrogen‐induced soil acidification in Chinese croplands. Global change biology 2020, 26, 3738-3751. https://doi.org/10.1111/gcb.15101"
We have deleted the tenth reference in the manuscript and replaced it with "Ren, C.; Jin, S.; Wu, Y.; Zhang, B.; Kanter, D.; Wu, B.; Xi, X.; Xin, Z.; De, C.; Jian, X.; Gu, B. Fertilizer overuse in Chinese smallholders due to lack of fixed inputs. Journal of Environmental Management 2021, 293, 112913. https://doi.org/10.1016/j.jenvman.2021.112913"
We have deleted the eleventh reference in the manuscript and replaced it with "Zhu, Q.; de Vries, W.; Liu, X.; Hao, T.; Zeng, M.; Shen, J.; Zhang, F. Enhanced acidification in Chinese croplands as derived from element budgets in the period 1980–2010. Science of the Total Environment 2018, 618, 1497-1505. https://doi.org/10.1016/j.scitotenv.2017.09.289"
Point 3: The vertical fonts in Figure 1 are recommended to be arranged horizontally at the top of the corresponding box, which is convenient for readers to read.
Response 3: Great idea! We have reworked Figure 1.
Point 4: Be careful, the map of China in Figure 2 lacks a scale bar. The expansion of the area does not require two lines, one line is OK.
Response 4: Great suggestion. We have revised Figure 2.
Point 5: It is detailed that the questionnaire survey sample is well representative, which is very important for social science research, otherwise the results will be biased. But what you said is a bit too detailed, the same sentence can also be changed. For example, multiple occurrences of ‘In order to’.
Response 5: Thanks for your precious comments. We have deleted the sentence "First, Liaoning province is divided into four major regions according to geographical location: east, west, south and north. Second, … Third, … Fourth, … In order to … The actual survey was conducted … certain extent" between lines 307-323 in the manuscript and inserted the following sentence between lines 380-390:
“First, Liaoning province was geographically divided into four major regions: east, west, south and north. Second, two administrative counties (cities) were randomly selected from within each geographic region. Third, 2-3 administrative villages were randomly selected from within each selected survey county (city). Fourth, the surveyed farmers were randomly selected within each selected administrative village. The team first conducted a small-scale pre-survey in Hengren County to ensure the rationality and rigor of the questionnaire. For meeting the authenticity and validity of the survey results, the investigators were trained in statistical methods and language expressions and other related aspects. The actual survey was conducted in the way of farmers' dictation and surveyors filling out the questionnaire on site, which ensured the integrity of the data to a certain extent.”
Point 6: It may be better to change Table 2 to a pie chart or a percentage histogram to enrich the look and feel of the article.
Response 6: Great idea! We have revised Table 2 to present it in the form of a pie chart (Figure 3).
Point 7: The discussion is not deep enough, and the discussion is to further enrich, demonstrate or extend the results on the basis of summarizing the literature. You are more self-analyzing than citing.
Response 7: We are grateful for your precious comments. After our careful literature review and discussion, the discussion section has been further improved. The sentence "Existing studies on farmers' technology adoption have focused their perspectives more on broad categories of agricultural …For example, …factors influencing pesticide technology preferences and finally verified the order of farmers' preferences and the main influencing factors under different technologies [74]" between lines 720-734 in the manuscript has been deleted and the following sentence has been inserted between lines 832-858:
“Existing studies on the use of STFFT among farmers have focused their perspectives more on influencing factors (external intervention elements, intra-individual characteristics), psychological decision-making processes, knowledge literacy and post-utility [14,16,68-71]. For example, Qi et al. conducted a study of 30 neighboring villages in Taojiang County using spatial measurement methods. They explored in depth the factors influencing farmers' use of friendly fertilizer application technologies by combining family surveys, farmland quality surveys, remote sensing images, and digital elevation models. The results showed that personal characteristics and topographic conditions influenced farmers' decisions, and that farmers with higher education levels were more likely to adopt new technologies [72]. Similar findings were obtained in this study that the higher the level of education received by farmers, the greater their knowledge base and the easier it is to adopt STFFT and apply it in practice. Wen used microsurvey data from 691 specialized apple farmers in Shandong and Shaanxi provinces to elucidate the effects of technical training and land operation size on STFFT adoption within the framework of the theory of planned behavior. The conclusions show that technical training experience significantly motivates farmers to use STFFT, and this motivating effect is more pronounced for the group of farmers with large-scale operations [73]. This study similarly verified the role of technology training and scale of operation in promoting farmers' technology adoption behavior and obtained the pathways of action based on this. Similarly, Xue et al. used a logit linear regression model based on data from a sample of 700 farmers in the Loess Plateau and Bohai Bay regions to reconfirm that technology training has a differential impact on different size groups of farmers [74].”
Correspondingly, we have replaced items 68-74 in the reference list with the updated literature. Once again, we would like to express our sincere gratitude for your suggestions.
Point 8: The conclusion is too long, too long, too long, shorten it to less than 400 words. This is not a conclusion, more like a review article.
Response 8: Thanks! To simplify the conclusion section, we have eliminated redundant text and limited the conclusion section to 385 characters. The sentence "This study constructs an analytical framework of…Third, these 10 salience factors are both independent and interrelated with each other, and together they form a structural system of factors influencing farmers' adoption of STFFT…and the training content should be easy to understand and close to practical application, so as to improve farmers' ability to master and understand the technology" between lines 817-894 in the manuscript has been removed and the following sentence has been inserted between lines 986-1034:
“Based on a survey of 402 maize farmers in Liaoning Province, this paper constructed an analytical framework of "external environmental stimuli-perceived characteristics-decision making behavior" and explored the factors influencing farmers' acceptance of STFFT through a logistic-ISM model. Three main conclusions are drawn below.
First, the surveyed farmers' decision-making behaviors regarding STFFT were consistent with the analytical framework of technology acceptance theory and had significant profit propensity and risk aversion. Five categories of factors, namely, basic personal characteristics, family characteristics, business characteristics, cognitive characteristics, and external environmental characteristics, have significant effects on farmers' technology adoption behavior.
Second, in general, the level of education is a deep root factor affecting farmers' absorption of new agricultural technologies, and the more educated farmers are more likely to adopt STFFT. the depth of knowledge accumulation enhances farmers' information acquisition ability, based on which the ratio of technical training and planting income is indirectly enhanced, which motivates farmers to generate technology adoption behavior.
Third, the influence of factors located under different strata on farmers' STFFT adoption behavior is heterogeneous with multiple realization paths. Specifically, they can be divided into three intrinsic mechanisms, briefly summarized as two paths that unfold based on educational attainment, and one path that extends based on production scale.
Based on the above findings, this study can generate the following policy recommendations. First, accelerate rural land transfer. At present, China is still dominated by small-scale forms of agricultural operations, which to a certain extent limits the effectiveness of agricultural technologies. Therefore, the government can appropriately develop large-scale production, thus enabling the diffusion of agricultural technologies based on a more suitable starting point. Second, strengthen farmers' perception of the usefulness of STFFT and actively achieve technology transformation and utilization. The government should increase the publicity effect of STFFT, so that farmers can under-stand the effect of the technology in improving crop yield and increasing income through the most intuitive way. This will enable farmers to gradually change their traditional fertilization concepts and establish a scientific fertilization knowledge system. Third, reduce the cost of technology adoption by farmers and strengthen STFFT training. The government should increase the support and financial subsidies for large-scale households and fertilizer producers. At the same time, technical extension departments should strengthen the organization of farmers to learn relevant knowledge, thus improving the STFFT adoption rate.”
Thanks for your constructive comments—which helps us a lot to improve the manuscript. We tried our best to address your concerns.

Round 2
Reviewer 1 Report
The authors did good job and addressed all issues risen by reviewers and provided valid answers with revision. The paper ca be published in current version.
Author Response
There is not the reviewer’s comment to response